# Health Benefits and Risks of Consuming Spices on the Example of Black Pepper and Cinnamon

**DOI:** 10.3390/foods11182746

**Published:** 2022-09-07

**Authors:** Joanna Newerli-Guz, Maria Śmiechowska

**Affiliations:** Department of Quality Management, Gdynia Maritime University, Morska 83, 81-225 Gdynia, Poland

**Keywords:** black pepper, cinnamon, health benefits and risks, antioxidant properties

## Abstract

The aim of this study is to present the benefits and risks associated with the consumption of black pepper and cinnamon, which are very popular spices in Poland. The article presents the current state of knowledge about health properties and possible dangers, such as liver damage, associated with their consumption. The experimental part presents the results of the research on the antioxidant properties against the DPPH radical, which was 80.85 ± 3.84–85.42 ± 2.34% for black pepper, and 55.52 ± 7.56–91.87 ± 2.93% for cinnamon. The total content of polyphenols in black pepper was 10.67 ± 1.30–32.13 ± 0.24 mg GAE/g, and in cinnamon 52.34 ± 0.96–94.71 ± 3.34 mg GAE/g. In addition, the content of piperine and pepper oil in black pepper was determined, as well as the content of coumarin in cinnamon. The content of piperine in the black pepper samples was in the range of 3.92 ± 0.35–9.23 ± 0.05%. The tested black pepper samples contained 0.89 ± 0.08–2.19 ± 0.15 mL/100 g d.m. of essential oil. The coumarin content in the cinnamon samples remained in the range of 1027.67 ± 50.36–4012.00 ± 79.57 mg/kg. Taking into account the content of coumarin in the tested cinnamon samples, it should be assumed that the majority of cinnamon available in Polish retail is *Cinnamomum cassia* (L.) J. Presl.

## 1. Introduction

Herbs and spices are very important in food technology, gastronomy and home cooking. Spices are most often defined as products of plant origin [1]. Initially their use was mainly combined with flavoring and improving the appearance of food. Then, it was pointed out that spices have preservative, antioxidant and antimicrobial properties, which significantly affect the shelf life and value of food [2]. Although the preservative properties of spices have been known to mankind since ancient times, nowadays, thanks to highly advanced analytical methods, we are learning about the compounds that are responsible for this action [3]. Health-protective properties are another benefit of the use of spices. Herbs and spices are carriers of numerous chemical compounds with health-improving properties that can provide potential protection against cardiovascular diseases, neurodegenerative diseases, diabetes type 2, and cancer [4]. However, we must remember the differences in terminology and we want to pay special attention to the difference between medicinal herbs, which are classified as medicinal products, and herbs, which are in turn food products.

Black pepper [*Piper nigrum* L.] and cinnamon [*Cinnamomum* spp.] are very popular spices in Poland. In 2020, imports of black pepper to Europe amounted to 80,000 tons, of which 6% was imported to Poland, and it was the 6th largest importer of black pepper in Europe. Over 75% of Polish pepper imports are whole grains. Part of the imported pepper is re-exported, and the estimated domestic consumption in 2020 was 5400 tons [5]. Among European countries, Poland ranks 6th when it comes to cinnamon imports (other than cinnamon Cinnamomum zeylanicum Blume neither crushed nor ground) and it accounts for 0.74% of total world imports. When it comes to cinnamon Cinnamomum zeylanicum B. neither crushed nor ground, Poland ranks 8th among the European countries with imports amounting to 0.32% [6]. Those two spices are one of the most consumed spices in Europe and Poland with a wide range of uses [7,8]. The aim of this work was to make a commodity characterization of these spices in the context of their potential importance for the consumer. In the above-mentioned spices, the main bioactive substances were determined and compared with the results of other authors’ research. Specifically, the antioxidant properties of the spices were determined as these are most often mentioned in the discussion of health-promoting properties of spices. Additionally, piperine and coumarine were chosen as representative biomarkers of the quality of black pepper and cinnamon, respectively. Piperine is the most important bioactive compound of black pepper, while coumarin due to its hepatotoxic activity should be closely monitored in any products with the addition of cinnamon.

An attempt was also made to determine the significance of the addition of a given spice to food on its health-promoting impact.

## 2. Characteristics of Selected Spices and Its Relevance to the Consumer

### 2.1. Black Pepper [Piper nigrum L.]—Origin, Types and Properties

Black pepper [*Piper nigrum* L.] is one of the most popular and one of the oldest spices [9]. It comes from the Malabar Coast of south-western India from where its journey around the world began. It is grown in many countries, including India, China, Indonesia, Malaysia, Brazil, Sri Lanka, and Vietnam. The great career of pepper began in the fifteenth and sixteenth centuries with the era of geographical discoveries. The spice is the dried seeds of this tropical evergreen plant. It occurs in several stages of harvesting as white, green and black pepper and differs in the degree of maturity and method of treatment.

Black pepper is obtained from unripe fruits by drying, most often in the sun, until the skin acquires a very dark color, almost black. Many factors influence the quality of black pepper (Figure 1). One of the factors affecting the quality of this spice is the variety of the plant. The Indian Spice Research Institute in India maintains the world’s largest collection of over 2000 varieties [9]. Other factors shaping the quality of black pepper are details of production including, inter alia, conditions of growing, harvesting, threshing, drying, cleaning, storage and standardization. The quality of black pepper depends largely on the size of the corns, color, the content of light, damaged peppercorns, humidity, the presence of foreign bodies such as animal excrement, and the presence of insects. These factors are fundamentally determined by the practices of harvesting, processing and the treatment of peppercorns on plantations and the classification and storage procedures adopted by exporters. Another aspect of quality is the level of microbial contamination, which should not exceed the permissible limits [10]. The commercial quality of black pepper is determined by pre-treatment and treatment of grains after harvest. Most often, peppercorn is blanched and intensively dried, as a result of which it usually contains less than 10% water. 

However, pepper drying is not always carried out in mechanical dryers. In India, a very popular method is the solarization of pepper, which involves drying pepper in the sun after blanching in hot water [11]. Pepper is grown in tropical conditions characterized by high temperature, high precipitation and humidity. Such conditions are favorable to the development of fungi, which leads to an increased occurrence of mycotoxins, especially in the absence of Good Agricultural Practices (GAP), Good Manufacturing Practices (GMP) and Good Hygiene Practices (GHP) [12]. For additional protection of black pepper against mycotoxins, as well as for decontamination, gamma irradiation is used [13]. Increasingly, however, consumers are looking for organic or bio-based products. Such alternative cultivation methods include, for example, the combined use of *Rhizosphere* bacteria with endophytic bacteria, which inhibits root diseases and increases the productivity of black pepper. Organic cultivation, on the other hand, provides pepper without pesticide and herbicide residues [14,15,16]. Additional factors affecting the quality of pepper are climate change and economic and social factors. Research conducted by Karamawati et al. [17] shows that Indonesian farmers are switching from pepper to oil palms farming for this reason, which brings higher profits and requires less work. The growing conditions and commercial quality of black pepper affect the sensory characteristics, which gives satisfaction to consumers. Unfortunately, the consumers are often disappointed with the quality of the spice when they discern the lack of information about the origin of the spice or the content does not correspond to the name of the product. To identify and study the authenticity of black pepper, very advanced research methods such as infrared spectroscopy with Fourier-transform using chemometry or metabolomic studies using ultra-high-performance liquid chromatography coupled with high-resolution mass spectrometry must be used [18,19]. These methods are very expensive, time-consuming and require a huge commitment of forces and resources and very well-equipped research laboratories. Therefore, the study of authenticity and traceability seeks fast, non-destructive methods based on the analysis of various data using chemometry, which will shorten and accelerate the research at least at the initial stage of identification [20,21]. The sensory and health properties of black pepper arise from its composition. Black pepper (*Piper nigrum* L.) contains compounds that can be classified into three groups. The first group consists of compounds that determine the sharpness of black pepper, the second includes substances that determine its aroma, and the third group consists of other compounds such as: fiber, starch, polyphenols, mineral salts, lipids [22]. The compound, which determines the pungent aroma of black pepper is piperine, together with its analogues.

From its detection by Hans Christian Ørsted in 1819, 55 piperine analogues were identified in black pepper [23]. The essential oil of black pepper also has very rich composition. It contains terpene and sesquiterpene hydrocarbons and their oxidized forms in the amount of 1–3%. Its composition is influenced by geographical origin and respective production process. A very important ingredient of black pepper is oleoresin, derived by extraction in the amount of 6–13%. It consists of 15–20% of essential oil and 35–55% piperine [24,25,26]. For the consumer, the quality of spice is determined by its seasoning properties (the smell and taste giving and improving in different dishes). The health properties of black pepper and piperine are presented on Figure 2 and in Table 1.

### 2.2. Cinnamon (Cinnamomum spp.)—Characteristics, Properties, Use 

Cinnamon, next to pepper, is one of the most widely used spices in the world. Cinnamon (*Cinnamomum* Scheffer) belongs to the *Lauracae* family, which includes more than 200 species. Cinnamon trees are evergreen trees growing in tropical climates in countries such as India, Indonesia, Philippines, Sri Lanka, Myanmar, Vietnam and China. However, of these, only a few have a unique significance in medicine, cosmetology and as a spice (Table 2).

Species of cinnamon trees differ in the morphological features of the leaves, such as shape, color, size and veining. Other distinguishing features of cinnamon trees are the morphological features of flowers, fruits and bark. The properties of cinnamon are mainly due to flowers, fruits, leaves and bark composition. The most valued part of cinnamon due to its unique healing properties is the bark of *C. verum* J. Presl (synonym of *C. zeylanicum* Blume) [76]. The bark of *C. verum* J. Presl is thin, softer than other cinnamon trees bark, paper and curls inwards on both sides, while the bark of *C. cassia* L. is hard, thicker and curls only on one side, see Figure 3. Individual species of cinnamon bark also have a different color, aroma, tenderness and taste.

The medicinal and spicy properties of *C. verum* J. Presl cinnamon bark result mainly from its composition. In addition, Ceylon cinnamon is more expensive due to its chemical composition, high quality, health benefits, and trace amounts of coumarin, which is found in higher concentrations in Cassia cinnamon. Table 3 shows the bark composition of two popular cinnamon trees.

A number of compounds contained in cinnamon such as aldehydes, alcohols, acids, esters, terpenes and others cause that it is used as a flavoring agent in seasonings, sauces, pastries and sweets, drinks, meat dishes, cereals, chewing gums and fruit preserves. The cinnamaldehyde in cinnamon is responsible for its sweet taste, and the addition of sugar in food products further enhances this effect. Cinnamon oil obtained from bark is also a flavoring agent in perfumes and toilet waters, lotions, shampoos, soaps and other cosmetics [82].

In the food trade, cinnamon occurs in the form of bark fragments of various sizes, but more often in the form of ground powder. Ground cinnamon is sometimes falsified and instead of *C. verum* contains *C. cassia* or powder of other cinnamon trees, and on the labels of food products containing cinnamon there is no information about the origin of the raw material [83]. For this purpose, very specialized and advanced analytical methods are used to detect adulteration, which allow the identification of the raw material [51,84,85,86,87].

The identification of geographical origin is also important due to the presence of coumarin in cinnamon. Coumarin compounds with a benzopyrone structure are found in plants of the families *Leguminosae* (Legumes), *Rutaceae* (Ruthaceae), *Umbellifereae* (Apiaceae) (Umbellate), *Compositae* (Complex), *Gramineae* (Grassy), forming connections of a glycosidic nature. However, in 1954, when the FDA report on the hepatotoxic effects of coumarin was published, the European Commission directive 88/388/EEC (Annex II) recommended that the intake of coumarin from natural sources, used as a food additive, should not exceed 2 mg/kg of food.

In 2005, Chemisches and Veterinäruntersuchungsamt in Münster, Germany, determined a coumarin content of 22 mg/kg in a sample of Christmas cinnamon cookies, which led to a discussion on increasing the supervision of food products containing coumarin, as well as on compliance with the limits by food producers [88]. As a result of these studies and discussions, the EFSA (European Food Safety Authority) has set the Acceptable Daily Intake (ADI for animals at 0.1 mg/kg body weight based on the maximum non-harmful dose (NOAEL). In 2008, the EFSA established the same TDI for humans [89].

*C. verum* is mainly used for medicinal purposes, due to the fact that only this species of cinnamon contains small amounts of coumarin. Cinnamon for medicinal purposes has been used since ancient times [90] and the medicinal properties of cinnamon have been confirmed in numerous studies (Figure 4 and Table 4).

## 3. Materials and Method

Samples (25) of black pepper (*Piper nigrum* L.) were obtained from the polish market. The test samples were representative of the entire Polish market. They came from a network of super and hyper stores present throughout the country, i.e., Auchan, Carrefour, Biedronka Jeronimo Martens, Kaufland, Lidl, Netto, Dino. These included both branded products and own brands of retail chains. All samples were encoded and analyzed. The piperine, essential oil content and antioxidant activity against DPPHꞏ radical and total polyphenols were investigated.

Material for the study consisted of 12 samples of ground cinnamon (*Cinnamomum* spp.) purchased in Poland. The coumarin content and antioxidant activity against DPPHꞏ radical and total polyphenols were investigated.

### 3.1. Used Chemicals

2,2-diphenyl-1-picrylhydrazyl (DPPH), Folin–Ciocalteu reagent, gallic acid, standard coumarin (purity ≥ 99%) were purchased from Sigma Aldrich GmbH (Steinheim, Germany), HPLC grade methanol and ethanol were obtained from POCH S.A. (Gliwice, Poland).

Five packages of each spice sample were taken from the market according to PN-ISO 948 [112], then milled in case of black pepper and cinnamon according to PN-ISO 2825 [113], and then 1% aqueous extracts were prepared.

### 3.2. Total Phenolic Content

The total phenolic content (TP) in water crude extracts was determined by the Folin–Ciocialteau method with modifications [114] 2.5 mL of 0.2 N FC reagent was added to tested solutions and mixed. After 5 min, 2 mL 75 g/L Na_2_CO_3_ solution was added. After 120 min incubation, the absorbance relative to that of a prepared blank was read at 760 nm using a spectrophotometer (Unicam UV2, Varian). The TP content is expressed in mg of gallic acid equivalents (mg GAE/g of product).

### 3.3. DPPH Free Radical Scavenging

Antioxidant activity was determined using the DPPH reagent and showed as DPPH radical scavenging percent. Free radical scavenging effect was determined using the free radical DPPH (2,2-diphenyl-1-picrylhydrazyl) reagent. 1 mL of the extract was added to 2 mL DPPH. The samples were gently mixed and left to stand in the darkness for 60 min. Absorbance was read at 517 nm using spectrophotometer. A control sample was prepared by mixing DPPH with distilled water. The ability of extracts to scavenge DPPH free radicals was calculated according to the following equation:$$\mathrm{Radical}\;\mathrm{scavenging}\;\mathrm{activity}\;[\%]\;\mathrm{AA}\left[\%\right]=\frac{{\mathrm{Abs}}_{\mathrm{contr}}-{\mathrm{Abs}}_{\mathrm{sample}}}{{\mathrm{Abs}}_{\mathrm{contr}}}\times 100$$

The values are presented as the means of triplicate analyses.

### 3.4. Piperine Assay

Crushed black pepper was extracted with ethanol (97% or absolute), and the obtained extracts were determined spectrophotometrically at a wavelength λ = 342 nm using UV-VIS spectrophotometer Unicam. The determination was made in three repetitions [115].

### 3.5. Essential Oil Content

The determination of essential oil content in crushed black pepper was made by steam distillation in a Dering apparatus [116]. The essential oil content was expressed as mL/100 g d.m. Determinations were performed in triplicate.

### 3.6. Coumarin Content

The separation of coumarin was obtained in ground cinnamon by the HPLC method using RP-Nova Pack C18 (240 × 4.6 mm, 5 µm) column. A mobile phase was composed of acetonitrile water with a gradient elution at a flow rate of 1 mL/min. The identification of coumarin in the cinnamon samples was conducted based on their retention time compared with the retention time of the standard. The HPLC method was validated in term of linearity, limit of detection, limit of quantification, precision and accuracy.

### 3.7. Statistical ANALYSIS

Results were presented as the mean and standard deviation. The experimental designs and calculations were conducted using the Software Package Statistica 10.0 (StatSoft Inc., Tulsa, OK, USA).

The experiments were evaluated using analysis of variance (ANOVA) to find the impact of the type and quantity of spices on the evaluated parameters. Statistical hypotheses were verified at a significance level of *p* values < 0.05.

## 4. Results and Discussion

The analyses of selected bioactive compounds content and antioxidant activity were carried out in samples of black pepper and cinnamon available on the Polish retail market. These studies will allow to assess the amounts of bioactive substances which Polish consumers will find in these spices. The obtained results were compared with the other authors’ studies.

### 4.1. Analysis of Black Pepper (Piper nigrum L.)

Black pepper samples (25) were encoded, the total content of polyphenols, antioxidant activity, the content of essential oil and piperine were determined. The analyses were performed in three repetitions and the results are presented in Table 5.

#### 4.1.1. Antioxidant Properties of Black Pepper (*Piper nigrum* L.)

The research showed different antioxidant activity of the tested black pepper samples available on the Polish market. The content of total polyphenols ranged from 9.75 to 32.13 mg GAE/g. The results of the statistical analysis confirmed that the pepper origin (brand) significantly influences the total content of polyphenols (KW-H(24) = 114.126, (*p* = 0.000)). The antiradical activity of the tested samples was at a high level, ranging from 63.93% to 85.42%. Statistical analysis showed that the brand of pepper influences the ability to scavenge DPPH free radicals (KW-H(24) = 114.63, *p* = 0.001).

The obtained results of our own research showed that the content of essential oils in pepper significantly influences the antioxidant activity measured with the use of DPPH radicals. A statistically significant, positive weak correlation between these parameters was found *r* = 0.251 (*p* = 0.045). It means that as the content of essential oil increases, the ability of the spice to scavenge DPPH radicals also increases. It should also be noted that pepper oil has a moderate ability to scavenge free radicals compared to other oils [117]. Gülçin points to the strong antioxidant properties of water and ethanolic black pepper extracts. The total content of polyphenols was determined at the level of 54.3 mg GAE in aqueous extracts [118]. The total content of polyphenols determined by Andradea and Ferreira, depending on the modification in the extraction, ranged from 14 to 27 mg GAE/g [119]. Ahmad determined the total content of polyphenols in the methanol extracts of black pepper purchased in Delhi, at the level of 172.8 mg GAE/100 g [120]. Nagy et al., determined the content of polyphenols in black pepper at the level of 338 ± 1.41 mg GAE/100 g, and the scavenging capacity of DPPH free radicals in methanolic solutions of pepper at the level of 13.28% [121].

The ability to scavenge free radicals of aqueous pepper extracts in the studies by Nahak and Sahu was for cubeb pepper (*Piper cubeba*) from 35.38 to 45.84% and for black pepper from 28.15 to 39.92%. In ethanol solutions it increased to 77.61% for cubeb pepper and 74.61% for black pepper [122]. The DPPH scavenging capacity of 35.20% was determined for an aqueous solution of black pepper by Gupta [123].

The polyphenol content of various pepper extracts was determined by Sruthi and Zachariah. Pepper extracts in chloroform and methanol showed the highest content of polyphenols compared to those in which n-hexane and water were used. In the case of black pepper, the total content of polyphenols in the aqueous solution was at the level of 3.84 µg GAE/g, and in the case of *P. longum* pepper, 2.16 µg GAE/g [124].

The high total polyphenol content in black pepper grown in Korea was determined by Lee et al., at the level of 1421.95 ± 22.35 mg GAE/100 g [125]. In another study, Lee et al., determined the total content of polyphenols in ethanol extracts of whole Korean peppercorns and after debarking, and it was 1046 ± 22 and 797 ± 28 mg GAE/100 g, respectively [26]. One of the lowest total polyphenols contents in black pepper from Bhubaneswar, India was shown by Mallick et al., and it was at the level of 11.9 ± 0.1 mg GAE/100 g [126].

In this context, the results obtained by Trifan et al., in different extracts are very interesting. The scavenging capacity of DPPH extracts with hexane, dichloromethane, 50% aqueous methanol and methanol was 18.77 ± 0.24, 19.56 ± 0.59, 45.41 ± 0.03, and 32.41 ± 0.07 mg Trolox equivalents (TE)/g, respectively. The presented research results indicate the importance of the analytical technique used in the extraction of the research material, and specifically in the type of polar and non-polar solvent used for the extraction of the spice [127].

In their research, Suchaj et al., showed a statistically significant effect of black pepper irradiation on the increase in antioxidant activity determined with the use of DPPH radicals after 2 months of storage. After 4 months of storage, these changes ranged from 4 to 9% [128].

#### 4.1.2. Piperine Content in Black Pepper (*Piper nigrum* L.)

The content of piperine in the studied samples of pepper varies. Three samples of pepper did not meet the requirements of the PN-A-86965:1997, the piperine content in them being less than the required 4%, but they met the requirements of Codex Alimentarius. A piperine content of 3.5% in black pepper is mandatory according to ESA 2015 and Codex Alimentarius FAO/WHO [129,130]. The content of piperine of four manufacturers was at the limit of the requirements of the standard and ranged from 3.92 to 3.98%. The remaining samples met the requirements of the standard, with the highest average piperine content equal 6.61%. The results of the statistical analysis confirmed that the brand of pepper affected the content of piperine. The calculated statistic was KW-H(24) = 620.83431 (*p* = 0.000).

The content of piperine at a similar level (2–7.4%) was determined by Ravidran [131]. Hamrapurkar et al., determined the piperine content of black pepper by the HPLC method. The piperine content was 8.13% in *P. nigrum* and 4.32% in *P. longum*, and these levels were slightly higher when extracted with CO_2_ gas (supercritical fluid) and methanol as a cosolvent and amounted to 8.76% and 4.96%, respectively [132]. A similar method was used by Rajopadhye et al., to determine the piperine content of various types of pepper purchased in Indian supermarkets. They obtained mixed results: 4.52% for *P. nigrum*, 3.71% for *P. longum*, and 1.19% for *P. cubeba* [133].

Zachariah et al., determined the piperine content at a much lower level from 2.8 to 3.8%. In a later study, the same authors determined piperine in black pepper from different locations in India: the lowest contents were found in pepper from Thevam (1.6%) and Neelamundi (2.0%), and the highest in samples from Perumkodi (9.5%) and Kuthiravally (8.7%) [134]. In addition to the type, the place of cultivation also affects the content of bioactive compounds in pepper. Sruthi et al., in the genus Panniyur-1, depending on the place of cultivation, determined the piperine content at the level of 2.13 to 4.49% [135].

The geographical origin also affects the varying levels of piperine content in black pepper. The high content of piperine in pepper from Malaysia (Malacca) was determined by HPLC by Rezvanian et al., and amounted to 5.85% [136]. Piperine in Indian and Malaysian pepper as determined by Jansz et al., was at the level of 2–7%, while in pepper from Sri Lanka above 7% [137]. Shango et al., found that the piperine content in black pepper depends on climatic factors such as air temperature, humidity, water availability and the amount of precipitation. In addition, they showed that the piperine content was influenced by the height at which the crop is located. Black pepper grown in Indonesia at an altitude of 650, 450 and 190 m above sea level contained 4.52%, 4.47% and 3.38% piperine, respectively [138]. The lowest piperine content in black pepper so far was determined by Ajaml [139]. The used reversed-phase liquid chromatography (RP-HPLC) method for the determination of piperine content in samples of pepper grown in various districts in Kerala, India, and it was in the range of 1.53 ± 0.002 to 1.78 ± 0.002 % *w/w*.

In turn, Lee et al. [125] determined piperine in samples of Korean pepper at the level of 2352.19 ± 68.88 mg/100 g. A similar content of piperine was determined by Shrestha et al. [140] in black pepper samples taken from various areas of Kathmandu in Nepal, and it ranged from 2.33% to 3.34% with an average of 2.75% and a standard deviation of 0.31%. In another study by Lee et al. [26] in Korean pepper samples, they determined piperine in the whole black pepper grain and in the grain with the peel stripped off, at 3728 ± 0.180 and 4035 ± 0.108%, respectively. It turned out that when the peel was removed, the piperine content was 8.2% higher, and the removal of the outer skin led to a softer taste and greater bioactivity of the black pepper.

#### 4.1.3. Essential Oil Content in Black Pepper (*Piper nigrum* L.)

While the content of piperine affects the sensory properties of pepper and determines its sharpness, the essential oil has a decisive influence on the aromatic properties. However, it should be remembered that both the content of piperine and essential oil are important factors influencing the health properties of black pepper, which is widely discussed in the theoretical part of this study. Moreover, the essential oil not only influences the aromatizing properties but also inhibits the growth of many microorganisms, including pathogenic microorganisms. The research of Nikoloć et al., showed that *P. niger* essential oil has a preservative effect on food [141].

The essential oil content of the 25 tested samples of black peppercorns was diversified. On average, it ranged from 0.89 to 2.19 mL/100 g of dry product. According to the requirements of the PN-A-86965 standard [117], the oil content should not be lower than 1.5 mL/100 g for black peppercorns and 1 mL/100 g for ground black pepper. The average content of essential oil in 50% of the tested samples did not meet the requirements of the above-mentioned standards, and was below 1.5 mL/100 g. According to Codex Alimentarius, the content of essential oil in black pepper should not be less than 2% of the dry weight [129,142].

The Kruskal–Wallis test showed the existence of differences in the essential oil content between the tested samples (KW-H(25) = 62.547, *p* = 0.001).

Published works on the content of essential oil in black pepper are rare and concern various aspects related to it. The content of essential oil, apart from factors related to the product, depends on the extraction conditions. The amount of essential oil increases as the extraction temperature increases. The efficiency of the process increases for black peppercorns from 0.4% after 40 min at 100 °C to 2.6% at a temperature of 250 °C [143]. These amounts are similar to those presented by Pino et al., who determined them at the level of 1 to 3% [144]. Rezvanian found the content of essential oil in peppercorns from Malaysia also at a relatively low level, ranging from 0.76% for Jahor pepper to 1.06% for Malacca pepper [138]. A more varied amount of essential oil is given by Zachariach et al. It ranges from 0.6% for the genus Perunkodi to 6.0% for the genus Subhakara [145].

Kurian et al., observed the variability in the essential oil content in the range from 2.7 to 5.1%, they also believe that the classical hydrodistillation method is the best method of obtaining volatile oils compared to other techniques [146]. On the other hand, other researchers publish results showing significantly lower levels of essential oil, also obtained by hydrodistillation. Rmili et al., confirmed that the hydrodistillation method is one of the best methods for obtaining essential oil from black pepper, however, the amount of essential oil obtained by them from black pepper remained at a low level of 1.24% [147]. A slightly higher content of essential oil in the range of 1.60–2.80% was obtained by hydrodistillation from black pepper by Hussain et al. [148].

The results of other authors’ studies confirm that the location of crops also significantly affects the content of essential oil in the same type of pepper. Sruthi determined the content of an oil of the Panniur-1 genus depending on the cultivation location, the determined amounts of the oil varied, and their content ranged from 1.6 to 3.2% [135]. Additionally, the studies carried by Chen et al., showed that the content of essential oil in pepper is influenced by the place of cultivation. They tested 25 samples of black and white pepper from different growing regions. The Investigated black pepper from the Yunnan Province in China contained 4.12%, and white pepper contained 3.01% of essential oil. On the other hand, the content of essential oil obtained from pepper originating in Indonesia and Vietnam was at the level of approximately 2% [25].

### 4.2. Analysis of Cinnamon (Cinnamonum *sp.*)

Table 6 presents the results of the determination of the content of bioactive substances and antioxidant properties in cinnamon samples purchased on the Polish retail market. They are referred to the other authors studies.

#### 4.2.1. Antioxidant Properties of Cinnamon (*Cinnamonum* sp.)

Among the analyzed samples, the highest total content of phenolic compounds was found in the sample 11 at 97.17 mg GAE/g, and the lowest in the sample 9 at 52.345 mg GAE/g (Table 6). The highest capacity of scavenging DPPH radicals was observed in the samples of producer 11 (91.87% on average), and the lowest in sample number 5, for which it amounted to 55.52%. The performed statistical analysis confirmed that the origin (brand) of cinnamon significantly influences the total phenolic content (K-W, H(15) = 59.025, *p* = 0.001), and the ability to scavenge free radicals (K-W, H(15) = 69.582, *p* = 0.000).

The total content of polyphenols in cinnamon bark determined by Abraham was 289.0 ± 2.2 mg GAE/g of plant [149].

The highest content of polyphenols was found in cassia bark ethanol extracts (9.534 g GAE/100 g d.m.), in leaves (8.854 g GAE/100 g d.m.) and the lowest in buds (6.313 g GAE/100 g d.m.). Extracts obtained from extraction with a CO_2_ supercritical fluid with metanol were characterized by a lower content of total polyphenols [150]. In addition, they confirmed the high DPPH radical scavenging capacity (over 80%) of *Cinnamomum cassia* ethanolic extracts. They obtained the highest values for cinnamon leaves.

Mathew and Abraham found a statistically significant decrease in the concentration of the DPPH radical along with an increase in the concentration of cinnamon bark extract from about 60% for a 3.125 µg/mL solution to about 5% for a 50 µg/mL solution [151]. Dragland found very high concentrations of antioxidants (>75 mmol/100 g) in the *Cinnamomi cortex* [152]. Additionally, other studies confirm the ability of plants from the *Cinnamomum* family to scavenge free radicals [153,154,155].

Prasad found a difference in the DPPH free radical scavenging ability between different grades of cinnamon. They decrease as follows: *C. zeylanica* > *C. cassia* > BHT > *C. pasiflorum* > *C. burmannii* > *C. tamala* [156]. The aqueous and alcoholic extracts of cinnamon (1:1) showed a significant ability to inhibit lipid oxidation in the in vitro lipid oxidation test [157].

Cinnamon has a higher oxidation inhibitory capacity than BHA, BHT and propyl gallate tested in the lipid peroxidation test [158]. Prakash et al. [159] found a high free radical scavenging capacity in solutions from cinnamon bark of *C. zeylanicum*, similar to the characteristics obtained by ascorbic acid.

Lin et al., assessed the antioxidant activity (using the DPPH radical) of 42 types of essential oils, including cinnamon oil. They showed the highest DPPH radical scavenging activity among the studied essential oils at the level of 91.4 ± 0.002% [160]. In another test, this spice was recognized as the best natural antioxidant, stronger than its synthetic counterparts (BHA, BHT). This is important to extend the shelf life of foodstuffs as oxidation is one of the most common chemical reactions responsible for food spoilage. It is likely that the high content of flavonoids is an essential basis for such a strong antioxidant activity [161].

Interesting, especially related to its use in cooking, is the fact that cinnamon’s antioxidant properties increase with the extraction temperature. Shobana and Akhilender Naidu subjected that the extract in a temperature of 100 °C for 30 min, not only did not lose its antioxidant properties, but rather showed a significant increase in them. “Cold” cinnamon extract had an antioxidant activity equal to 21%, while when cooked it increased by as much as 35% [157].

Trifan et al., showed that the antioxidant properties of cinnamon depend on the extraction method. They extracted the cinnamon bark with hexane, dichloromethane, 50% aqueous methanol and methanol. The result was a total polyphenol content of 18.46 ± 0.27, 14.00 ± 0.15, 92.90 ± 0.46, and 63.68 ± 1.48 mg GAE/g, respectively. In the same study, the antioxidant activity (using the DPPH radical) was as follows: 6.96 ± 0.41, 9.39 ± 0.57, 473.74 ± 1.45, 178.42 ± 0.81%. These results clearly indicate a high relationship between antioxidant activity and the extraction technique, i.e., the polarity of the solvent, which influences the extraction of cinnamon essential oil [127].

#### 4.2.2. Coumarin Content in Cinnamon (*Cinnamonum* spp.)

In 16 samples of cinnamon purchased on the Polish market, the varied content of coumarin was determined and ranged from 1027.67 ± 50.36 to 4012.00 ± 79.57 mg/kg. The performed statistical analysis allowed us to establish the existence of a relationship between the origin (brand) of cinnamon and the content of coumarin (K-W, H(15) = 35.325, *p* = 0.0003). The conducted research allows us to assume due to the high content of coumarin, but also a strong aroma and slightly sweet taste that we are dealing with cassia.

These results are also confirmed by other authors [162,163,164]. Authors examining the content of coumarin in cinnamon found that it is directly related to the species of cinnamon and the degree of processing (it differs for the bark of cinnamon and ground cinnamon). Woehrlin et al., report the coumarin content in cinnamon bark at the level of 1740 to 7670 mg/kg [165]. Its content in the bark found in Chinese studies was even over 12,000 mg/kg [166]. In ground cinnamon, Blahova and Svobodova [164] gave it at the level from 2650 to 7017 mg/kg in samples from the Czech market. A high level of coumarin was also determined by Ho et al.—29,400 mg/kg DS [167]. On the other hand, Lungarini found very low contents (<100 mg/kg) in ground *C. verum*, and up to 3094 mg/kg in cassia [168].

The higher content of coumarin in cassia than in Ceylon cinnamon is also confirmed by studies by other authors. Sproll et al. [163] found in the tested samples from the German market the absence of coumarin in Ceylon cinnamon, while in cassia, it was determined at the level of 2880–4820 mg/kg. In the samples of cinnamon of undefined origin, these amounts reached even 8790 mg/kg, and 85% of the 20 samples contained coumarin, so it can be presumed that 15% were samples of Ceylon cinnamon.

Among the food products containing cinnamon, an important source of coumarin were cinnamon cookies with up to 88 mg of coumarin/kg, and bread and breakfast cereals (up to 32 mg of coumarin/kg). This results in an amount exceeding the TDI by a child consuming just three to four cinnamon cookies weighing 5 g, and for an adult it is about ten pieces. Coumarin was present in small amounts in dairy products (up to 2 mg/k) and alcohols (up to 8 mg/kg) [163].

Research conducted in India has shown that cinnamon sourced directly from plantations has a low to moderate coumarin content, ranging from 12.3 to 143 mg/kg. On the other hand, a high content of coumarin was found in samples of ground cinnamon purchased in retail stores and it ranged from 819 to 3462 mg/kg. Only one market sample of cinnamon purchased from an authentic spice vendor showed low coumarin content (19.6 mg/kg). The authors of the study suggest that market cinnamon is very often adulterated with cheaper substitutes such as *C. cassia* and *C. burmanii*, because market cinnamon containing the least coumarin was three times more expensive than other market products [169].

In spice mixtures containing cinnamon, the coumarin content reaches 4308 mg/kg [170], and in plants from the *Lamiaceace* family it ranges from 14,300 to 276,900 mg/kg DS, *Lavandula*-lavender and *Salvia*-sage show its highest content [171].

Authors including Sproll [163] and Abraham [149] consider that regulations on the content of coumarin in food should be established, which was reflected in the Regulation of the European Parliament and of the Council (EC) No. 1334/2008 of 16 December 2008 [172].

Ground cinnamon and spice blends purchased in retail stores have led food regulatory authorities in many countries to increase the frequency of inspections [173,174]. The Norwegian Scientific Committee for Food Safety (Vitenskapskomiteen for mattrygghet, VKM), at the request of the Norwegian Food Safety Authority (Mattilsynet), conducted a risk assessment of coumarin consumption in the Norwegian population. As a result of these studies, it was found that a small or occasional exceeding of the TDI is not considered to increase the risk of adverse health effects. The consumption of coumarin may in some cases exceed the TDI seven to twenty times. Liver toxicity may occur soon after the initiation of coumarin exposure. Such large daily exceedances of TDI, even within a limited period of 1–2 weeks, raise concerns about adverse health effects [175].

## 5. Conclusions

The principal role of spices is to raise the quality of food to a higher level. Hence, spices are used primarily to provide consumers with a sensory experience and pleasure from consuming food [176]. Black pepper (*Piper nigrum* L.) and cinnamon (*Cinnamomum* spp.) are frequently consumed spices in Poland. The research discussed in this paper shows that these spices, apart from influencing the sensory value of food, may have a positive effect on the human body. The research carried out in this study shows that the quality of black pepper available to consumers in retail trade was at a good level, as measured by the results of the content of piperine and pepper oil. The content of piperine in the black pepper samples was in the range of 3.92 ± 0.35–9.23 ± 0.05%. The tested black pepper samples contained 0.89 ± 0.08–2.19 ± 0.15 mL/100 g d.m. essential oil. The coumarin content in the cinnamon samples remained in the range of 1027.67 ± 50.36–4012.00 ± 79.57 mg/kg. Taking into account the coumarin content, we suppose that the majority of cinnamon available in the Polish retail trade is *Cinnamomum cassia* (L.) J. Presl.

In the wider context of the research carried out, the question is what are the benefits of consuming spices for consumers? These benefits appear to depend primarily on the amount of spices consumed and their quality, measured by the content of the important bioactive substances. Therefore, the inspection services face a major task of ensuring that the quality of spices available on the market is as high as possible, because only then will these spices will have a high sensory value and will have an impact on human health. Consequently, there is still a need for more research into how spices affect the specific organs of the human body [177,178].

## Figures and Tables

**Figure 1 foods-11-02746-f001:**
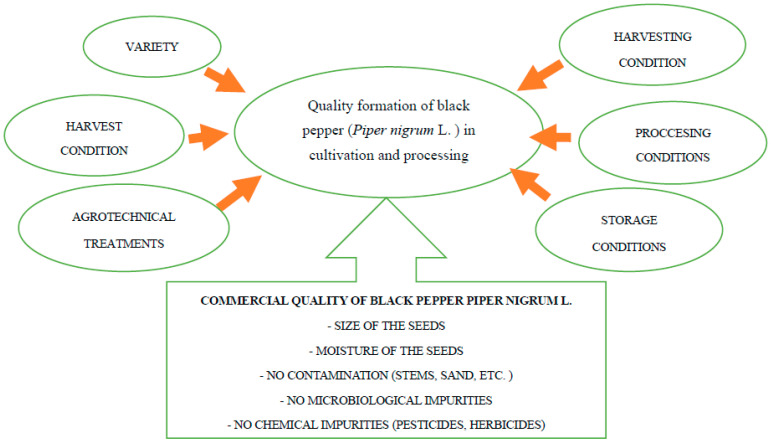
Factors influencing the black pepper (*Piper nigrum* L.) quality. *Source: own study*.

**Figure 2 foods-11-02746-f002:**
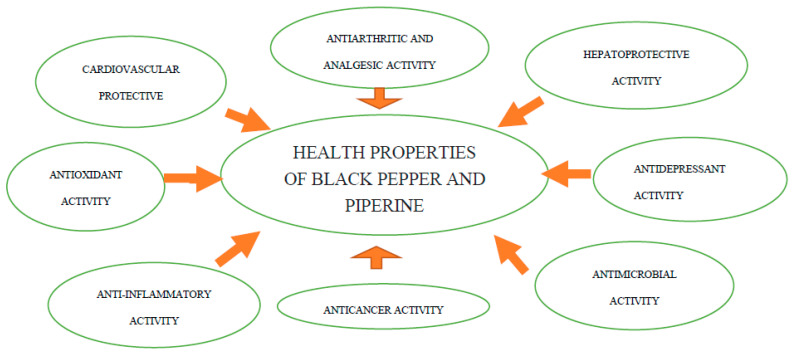
Health properties of black pepper and piperine. *Source: own study*.

**Figure 3 foods-11-02746-f003:**
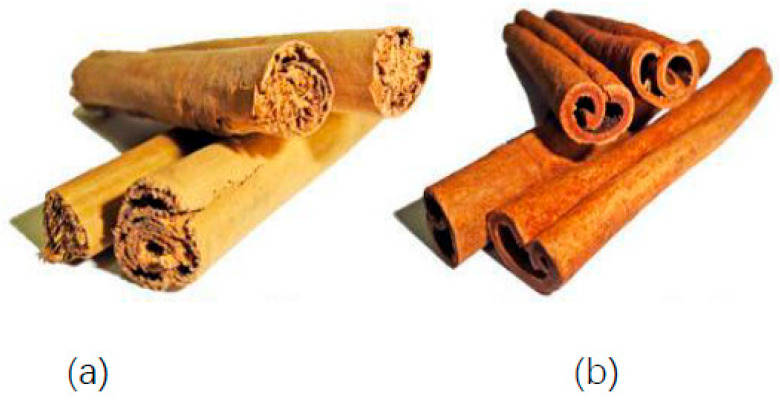
Cinnamon bark (**a**) *C. verum* J. Presl and (**b**) *C. cassia* L. *Source: [80]*.

**Figure 4 foods-11-02746-f004:**
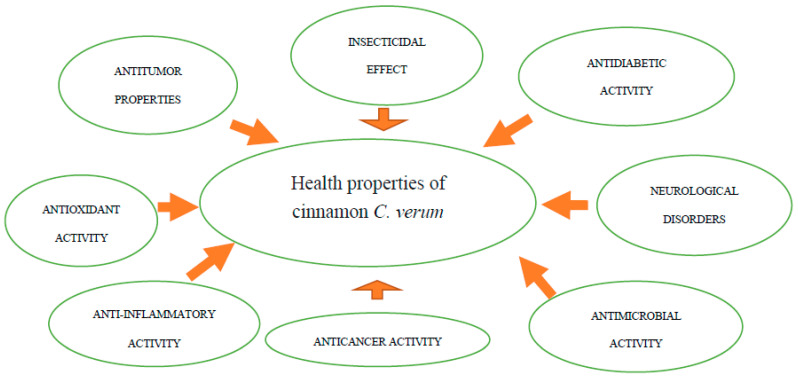
Health properties of cinnamon *C. verum*. *Source: own study*.

**Table 1 foods-11-02746-t001:** Health properties of black pepper and piperine based on other authors’ studies.

Health Effects	Source
Antiallergic effect	Aswar et al., 2015; Kim & Lee 2009 [27,28]
Analgetic activity	Bukhari et al., 2013; Jeena et al., 2014; Tasleem et al., 2014 [29,30,31]
Antiarthritic activity	Bang et al., 2009; Umar et al., 2013 [32,33]
Anticancer activity	Banerjee et al., 2021; Greenshields et al., 2015; Morsy & Abd El-Salam 2017; Prashant et al., 2017 [34,35,36,37]
Antidiabetic activity	Atal et al., 2016; Kharbanda et al., 2016; Oboh et al., 2013; Sarfraz et al., 2021 [38,39,40,41]
Antidepressant activity	Emon et al., 2020; Ghosh et al., 2021; Hritcu et al., 2015; Mao et al., 2014 [42,43,44,45]
Antihypertensive activity	Hlavačková et al., 2011; Lee et al., 2015; Taqvi et al., 2008; [46,47,48]
Antiinflammatory activity	Bang et al., 2009; Jeena et al., 2014; Tasleem et al., 2014; Yu et al., 2020 [30,31,32,49]
Antimicrobial activity	Bawazeer et al., 2022; Chen et al., 2019; Daigham & Mahfouz 2020; Hien & Dao 2022; Martinelli et al., 2017; Morsy & Abd El-Salam 2017; Zhang et al., 2017; Zou et al., 2015 [36,50,51,52,53,54,55,56]
Antineurodegenerative activity	Chonpathompikunlert 2010; Elnaggar et al., 2015; Etman et al., 2018 [57,58,59]
Antiobesity activity	Du et al., 2020; Lailiyah et al., 2021; Meriga et al., 2017; Shah et al., 2011 [60,61,62,63]
Antioxidant activity	Jeena et al., 2014; Srinivasan 2007; Vijayakumar & Nalini 2006 [30,64,65]
Cardiovascular protection	Dutta et al., 2014; Taqvi et al., 2008; Wang et al., 2021 [48,66,67]
Gastrointestinal and antidiarrheal activity	Mehmood & Gilani 2010; Srinivasan 2007; Shamkuwar et al., 2012; Shamkuwar 2013 [64,68,69,70]
Hepatoprotective and pancreatitis activity	Bae et al., 2011; Christina et al., 2006; Gurumurthy et al., 2012; Matsuda et al., 2008; Nirwane & Bapat 2011 [71,72,73,74,75]

*Source: own study.*

**Table 2 foods-11-02746-t002:** Main varieties of cinnamon for medicinal and seasoning purposes.

Systematic Term	Synonym	Common Term	Occurrence	Source
*Cinnamomum verum* J. Presl	*Cinnamomum zeylanicum* Blume	Ceylon cinnamon True cinnamon	Sri Lanca	Weerasekera et al., 2021 [76]
*Cinnamomum cassia* (L.) J.Presl	*Cinnamomum aromaticum* Ness	Chinese cinnamon	China, North-east asia	Zhang et al., 2017 [55]
*Cinnamomum burmannii* (Nees & T.Nees) Blume	—	Indonesian cinnamon Java cinnamon	Indonesia, Wietnam, Filipines	Al-Dhubiab 2012 [77]
*Cinnamomum tamala* (Buch.-Ham.) T.Ness & C.H. Eberm.	*Cinnamomum albiflorum* Ness *Cinnamomum lindleyi* Lukman *Laurus tamala* Buch.-Ham.	Indian cassia Indian bay leaf Tejpatta	India, Nepal, Bhutan, China	Tiwari & Talreja 2020 [78]
*Cinnamomum loureiroi* Ness	—	Saigon cinnamon Vietnamese cinnamon	Wietnam	Kumar et al., 2019 [79]

*Source: own study.*

**Table 3 foods-11-02746-t003:** Basic composition of the bark of *Cinnamomum verum* J. Presl and *Cinnamomum cassia* L.

Cinnamon	Composition
*Cinnamomum verum*	cinnamaldehyde 1.99% (1.49–3.20%), cinnamylacetate, cinnamon alcohol 0.043% (nd–0.083%), eugenol, coumarin (nd–0.004%)
*Cinnamomum cassia*	cinnamaldehyde (0.005–9.383%), cinnamic acid (0.001–0.191%), cinnamyl alcohol (0.001–0.177%) cinnamamyl acetate, cinnamon alcohol, eugenol, coumarin 0.001–1.218% (up to 5%)

*Source: [81].*

**Table 4 foods-11-02746-t004:** Health properties of cinnamon based on other authors’ studies.

Health Properties	Source
Analgetic activity	Pandey & Chandra 2015 [91]
Anticancer activity	Dutta & Chakraborty 2018; Thompson et al., 2019 [92,93]
Antidiabetes activity	Anderson et al., 2015; Zare et al., 2019; Shinjyo et al., 2020 [94,95,96]
Antiinflammatory activity	Han & Parker 2017; Schink et al., 2018; Shishehbor et al., 2018 [97,98,99]
Antimicrobial activity	Abd El-Hack et al., 2020; Arancibia et al., 2014; Atki et al., 2019 [100,101,102]
Antiobesity activity	Mnafgui et al., 2015 [103]
Antioxidant activity	Gulcin et al., 2019; Weerasekera et al., 2021 [76,104]
Cardiovascular protection	Jain et al., 2017; Mnafgui et al., 2015; Mousavi et al., 2020; Tarkhan et al., 2019 [103,105,106,107]
Insecticidal effect	Khan et al., 2020; Attia et al., 2020 [108,109]
Neurological disorders	Kang et al., 2016; Khasnavis & Pahan 2014 [110,111]

*Source: own study.*

**Table 5 foods-11-02746-t005:** Antioxidant activity (TP, AA), piperine and essential oil content in black pepper.

	n	TP M ± SD [mg GAE/g]	AA_DPPH_ M [%]	Piperine M ± SD [%]	Essential Oil M ± SD [mL/100 g d.m.]
1	3	13.02 ± 1.56	84.42 ± 2.32	6.49 ± 0.07	1.09 ± 0.09
2	3	32.13 ± 0.24	84.51 ± 3.92	5.26 ± 0.03	1.29 ± 0.01
3	3	11.52 ± 0.30	83.93 ± 1.29	7.11 ± 0.05	1.77 ± 0.08
4	3	13.18 ± 0.35	85.42 ± 2.34	9.23 ± 0.05	2.05 ± 0.17
5	3	11.72 ± 0.85	84.86 ± 3.79	6.19 ± 0.05	1.12 ± 0.01
6	3	11.90 ± 0.29	82.67 ± 3.99	5.77 ± 0.04	1.48 ± 0.08
7	3	10.67 ± 1.30	85.32 ± 4.71	6.40 ± 0.05	1.31 ± 0.08
8	3	14.41 ± 0.73	81.49 ± 2.22	5.91 ± 0.00	2.08 ± 0.08
9	3	11.37 ± 0.18	82.01 ± 6.48	6.17 ± 0.04	1.43 ± 0.00
10	3	11.80 ± 1.13	80.64 ± 3.21	6.61 ± 0.02	1.03 ± 0.16
11	3	12.29 ± 0.78	84.27 ± 4.43	7.19 ± 0.05	2.18 ± 0.08
12	3	11.18 ± 0.06	80.85 ± 3.84	5.77 ± 0.04	1.75 ± 0.08
13	3	10.88 ± 0.37	85.13 ± 2.93	7.19 ± 0.05	1.55 ± 0.14
14	3	13.31 ± 0.12	81.73 ± 5.11	5.90 ± 0.06	1.32 ± 0.17
15	3	12.38 ± 0.54	78.07 ± 3.27	6.44 ± 0.03	1.86 ± 0.14
16	3	12.85 ± 1.02	84.45 ± 4.12	3.92 ± 0.35	1.61 ± 0.16
17	3	11.59 ± 0.27	71.29 ± 3.84	4.87 ± 0.02	1.48 ± 0.08
18	3	12.03 ± 0.32	81.83 ± 4.01	5.55 ± 0.02	1.42 ± 0.07
19	3	11.65 ± 0.70	84.60 ± 5.82	3.98 ± 0.06	2.17 ± 0.14
20	3	14.02 ± 1.08	85.34 ± 4.17	5.78 ± 0.64	1.47 ± 0.08
21	3	12.85 ± 1.02	84.43 ± 2.22	4.61 ± 0.12	1.76 ± 0.08
22	3	11.24 ± 0.53	84.59 ± 2.81	4.95 ± 0.17	1.28 ± 0.14
23	3	9.75 ± 0.96	73.88 ± 3.23	4.87 ± 0.02	0.89 ± 0.08
24	3	14.53 ± 0.61	81.48 ± 1.11	5.58 ± 0.01	1.48 ± 0.08
25	3	11.28 ± 0.74	63.93 ± 0.86	4.51 ± 0.22	2.19 ± 0.15

*Source: own study.*

**Table 6 foods-11-02746-t006:** Antioxidant activity (TP, AA) and coumarine content in cinnamon *Cinamomum* sp.

	n	TP M ± SD [mgGAE/g]	AA_DPPH_ M [%]	Coumarine M ± SD [mg/kg]
1	3	94.71 ± 3.34	90.48 ± 4.13	1027.67 ± 50.36
2	3	92.32 ± 2.01	69.39 ± 2.88	3120.67 ± 81.25
3	3	85.77± 1.29	88.285 ± 1.49	2107.33 ± 103.24
4	3	74.58 ± 5.71	87.57 ± 2.11	3157.67 ± 59.77
5	3	73.38 ± 3.82	55.52 ± 7.56	2240.67 ± 57.01
6	3	72.33 ± 5.53	85.50 ± 2.65	3383.67 ± 57.07
7	3	71.22 ± 1.16	82.53 ± 1.99	2368.33 ± 37.58
8	3	77.575 ± 1.59	72.03 ± 3.47	3111.12 ± 34.28
9	3	52.345 ± 0.96	68.19 ± 3.47	3564.67 ± 42.12
10	3	69.16 ± 1.24	76.12 ± 2.81	3066.33 ± 60.12
11	3	97.17 ± 2.18	91.87 ± 2.93	2725.32 ± 41.29
12	3	94.64 ± 2.36	72.64 ± 3.02	4012.00 ± 79.57
13	3	64.38 ± 1.67	90.48 ± 3.33	3198.71 ± 81.82
14	3	88.27 ± 1.11	79.94 ± 4.61	3798.00 ± 90.04
15	3	79.25 ± 2.13	91.11 ± 2,63	3812.24 ± 61.29
16	3	69.91 ± 1.18	87.51 ± 1.85	2603.77 ± 83.64

*Source: own study*

## Data Availability

The data are available from the corresponding author on request.

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
