# Peer review of "Health Benefits and Risks of Consuming Spices on the Example of Black Pepper and Cinnamon"

_foods, 2022, doi:10.3390/foods11182746_

Round 1

Reviewer 1 Report

1-These two spices could have been given in the title where two spices were examined. I think it is wrong to call it "chosen spice".

2-In the summary, a general statement was given about black pepper and cinnamon, and it was a bit of a slur. For example, what you call danger could be given as an example.

3-In the summary, it was said that coumarin and piperine were examined, which could explain why they were specifically chosen.

4- In the last sentence given in the summary, the expression to assume is strange.

5- The introduction part was the part where the authors expressed their thoughts, in this part scientific information about the subject could be shared.

6-Cinnamon and black pepper were chosen from the most consumed spices in Europe and Poland. It was targeted because piperine in black pepper and coumarin in cinnamon are important components. This information should have been expressed more concisely in the summary and introduction.

7-This work is like student work. Frankly, I didn't find it very professional.

8- If it were me, I'd add everything about the piperine in black pepper. I would look at what its chemistry is, how it was obtained, what its effects on health were, and who did what on it. Likewise, I would give the coumarin in cinnamon comprehensively. Because, depending on these substances in these selected spices, the benefits and harms on health should be compatible with the title. So this question comes to my mind. Do the good or bad effects on health depend only on these substances?

9- Overall it's a good job. For example, while black pepper causes allergies, does white pepper also cause allergies? White pepper is not well known in the world. This issue was on my mind, so such literature discoveries are also important.

10- In the conclusion part, similar assumptions and questions were added to the introduction. Again this struck me as strange.

Author Response

Answer to Comments and Suggestions for Authors by Reviewer 1

1-These two spices could have been given in the title where two spices were examined. I think it is wrong to call it "chosen spice".

Author Reply: We changed the title of the work to mention black pepper and cinnamon explicitly.

2-In the summary, a general statement was given about black pepper and cinnamon, and it was a bit of a slur. For example, what you call danger could be given as an example.

Author Reply: We mentioned specifically liver damage in the abstract as the prime example of the possible dangers. The hepatotoxic effects of coumarin exposure are the most important ones for the heavy cinnamon consumers.

3-In the summary, it was said that coumarin and piperine were examined, which could explain why they were specifically chosen.

Author Reply: The justification for the studied compounds is now provided in the main text at the end of Introduction. We think that abstract should be concise and longer justifications may be relegated to the main part of the work.

4- In the last sentence given in the summary, the expression to assume is strange.

Author Reply: We changed the expression to “suppose”, since our measurements serve only as an indirect indication of the botanical variety of cinnamon.

5- The introduction part was the part where the authors expressed their thoughts, in this part scientific information about the subject could be shared.

Author Reply: We shortened the Introduction and focused more on the justification of why the two specific spices were chosen for study.

6-Cinnamon and black pepper were chosen from the most consumed spices in Europe and Poland. It was targeted because piperine in black pepper and coumarin in cinnamon are important components. This information should have been expressed more concisely in the summary and introduction.

Author Reply: Please see the response to the previous question.

7-This work is like student work. Frankly, I didn't find it very professional.

Author Reply: In this revision, we tried our best to make our work sound more professional.

8- If it were me, I'd add everything about the piperine in black pepper. I would look at what its chemistry is, how it was obtained, what its effects on health were, and who did what on it. Likewise, I would give the coumarin in cinnamon comprehensively. Because, depending on these substances in these selected spices, the benefits and harms on health should be compatible with the title. So this question comes to my mind. Do the good or bad effects on health depend only on these substances?

Author Reply: The properties of the two studied spices are gathered in sections 2.1 and 2.2 for black pepper and cinnamon, respectively. Please see tables and figures in these sections for a concise summary of the relevant properties.

9- Overall it's a good job. For example, while black pepper causes allergies, does white pepper also cause allergies? White pepper is not well known in the world. This issue was on my mind, so such literature discoveries are also important.

Author Reply: Please be aware that white pepper was not the topic of this work. The reason is already given by the Reviewer, as black pepper is one of the most important spices on the Polish market and white pepper’s consumption is marginal in comparison. However, the possibility to cause allergies is not limited to black pepper and white pepper is able to cause similar adverse effects in humans. Please see for example https://pubmed.ncbi.nlm.nih.gov/14635474/ as a case study of cross allergy to black and white pepper. However, we chose not to include anything on white pepper in our work in order to not confuse the potential reader by its sudden inclusion.

10- In the conclusion part, similar assumptions and questions were added to the introduction. Again this struck me as strange.

Author Reply: We slightly changed the wording of the Conclusions section. We wanted to emphasize more general consequences of ensuring the quality of spices available to the consumers. 

Reviewer 2 Report

Review report on the manuscript "Health benefits and risks of selected spices consumption"
This work aimed to present the benefits and risks associated with the consumption of spices such as black pepper and cinnamon. However, the work shows several issues requiring attention of the authors. 

The following are major points for consideration of the authors:

1. The introduction part is too long and redundant. The important scientific issue should be highlight in a concise manner.

2. The introduction describes that the variety, origin, processing methods may affect the chemical quality of the materials (Black pepper and cinnamon). However, the collected samples in this article do not describe those information in detail. The collected samples should be able to cover the market characteristics of the whole Polish region (Line268-273).

3. The use of spectrophotometric measurements, such as total phenolic content (TP), antioxidant (DPPH free radical scavenging), and Piperine assay methods that employ chemical assays are accepted as screening methods. HPLC-based methods would bring about a more in-depth discussion on the findings (Line281-301).

The following are minor points for consideration of the authors, and please make a clear statement:

1. The title of this manuscript should be more specific. There are two spices of black pepper and cinnamon in this research, and should be indicated in title.

2. Essential oil content in black pepper (Piper nigrum L.), line 332, 382, 430; Antioxidant properties of cinnamon (Cinnamonum sp.), line484. Italics should be used.

3. p = 0,000. Italics should be used in all this manuscript.

4. many references are citied too much, and more 10-15 years ago.

5. why choose these two spices in this study, and just compare the difference of some actives?

Author Response

Answer to Comments and Suggestions for Authors by Reviewer 2

The following are major points for consideration of the authors:

  1. The introduction part is too long and redundant. The important scientific issue should be highlight in a concise manner.

Author Reply: We streamlined and shortened the "Introduction” section. We also deleted the entire short section 2 “Characteristics of spices and their importance for the consumer” which served just as the extension of introduction in order to focus more on the two studied spices.

  1. The introduction describes that the variety, origin, processing methods may affect the chemical quality of the materials (Black pepper and cinnamon). However, the collected samples in this article do not describe those information in detail. The collected samples should be able to cover the market characteristics of the whole Polish region (Line 268-273).

Author Reply: We added relevant data to the section mentioned above. Specifically, we chose the most important retail networks on the Polish market and purchased both products by own brands of the retail networks, as well as products of known brands. In this way, we obtained a wide market coverage from the point of view of an average Polish consumer.

  1. The use of spectrophotometric measurements, such as total phenolic content (TP), antioxidant (DPPH free radical scavenging), and Piperine assay methods that employ chemical assays are accepted as screening methods. HPLC-based methods would bring about a more in-depth discussion on the findings (Line 281-301).

Author Reply: The reviewer is certainly right about the advantages of using HPLC analysis, especially for the possible speciation of polyphenols present in the samples. However, this would require a major extension of the present work, for example providing the appropriate phenol standards. TP is still a concise and widely used method of characterizing the bioactive polyphenols in food products.

The following are minor points for consideration of the authors, and please make a clear statement:

  1. The title of this manuscript should be more specific. There are two spices of black pepper and cinnamon in this research, and should be indicated in title.

Author Reply: We changed the title of the work to mention black pepper and cinnamon explicitly.

  1. Essential oil content in black pepper (Piper nigrum L.), line 332, 382, 430; Antioxidant properties of cinnamon (Cinnamonum sp.), line 484. Italics should be used.

Author Reply: We corrected the style of the text.

  1. p = 0,000. Italics should be used in all this manuscript.

Author Reply: We corrected the style of the text.

  1. many references are citied too much, and more 10-15 years ago.

Author Reply: Some of the older references (such as EU directives) are impossible to be removed, however, we updated the list of references to include more recent studies.

  1. why choose these two spices in this study, and just compare the difference of some actives?

Author Reply: The two studied spices are one of the most widely used in Poland and Poland shares a wide portion of the European market in their imports. This is now underlined more clearly in the Introduction.

Round 2

Reviewer 2 Report

I carefully read this revision, it had been changed and made a great improvement, it could be accepted  for publication.